# Analysis of Single- and Double-Stranded DNA Damage in Osteoblastic Cells after Hyperbaric Oxygen Exposure

**DOI:** 10.3390/antiox12040851

**Published:** 2023-04-01

**Authors:** Nele Schönrock, Frauke Tillmans, Susanne Sebens, Wataru Kähler, Sebastian Klapa, Bente Rieger, Harry Scherthan, Andreas Koch

**Affiliations:** 1Naval Institute for Maritime Medicine, 24119 Kronshagen, Germany; 2Divers Alert Network, Durham, NC 27705, USA; 3Institute for Experimental Cancer Research, Christian-Albrechts-University, 24118 Kiel, Germany; 4Bundeswehr Institute of Radiobiology Affiliated to the University of Ulm, 80937 Munich, Germany

**Keywords:** oxidative stress, DNA damage, osteoblasts, gamma-H2AX, hyperbaric hyperoxia, comet assay, DSBs

## Abstract

(1) Background: Hyperbaric oxygen (HBO) exposure induces oxidative stress that may lead to DNA damage, which has been observed in human peripheral blood lymphocytes or non-human cells. Here, we investigated the impact of hyperbaric conditions on two human osteoblastic cell lines: primary human osteoblasts, HOBs, and the osteogenic tumor cell line SAOS-2. (2) Methods: Cells were exposed to HBO in an experimental hyperbaric chamber (4 ATA, 100% oxygen, 37 °C, and 4 h) or sham-exposed (1 ATA, air, 37 °C, and 4 h). DNA damage was examined before, directly after, and 24 h after exposure with an alkaline comet assay and detection of γH2AX+53BP1 colocalizing double-strand break (DSB) foci and apoptosis. The gene expression of TGFß-1, HO-1, and NQO1, involved in antioxidative functions, was measured with qRT-PCR. (3) Results: The alkaline comet assay showed significantly elevated levels of DNA damage in both cell lines after 4 h of HBO, while the DSB foci were similar to sham. γH2AX analysis indicated a slight increase in apoptosis in both cell lines. The increased expression of HO-1 in HOB and SAOS-2 directly after exposure suggested the induction of an antioxidative response in these cells. Additionally, the expression of TGF-ß1 was negatively affected in HOB cells 4 h after exposure. (4) Conclusions: in summary, this study indicates that osteoblastic cells are sensitive to the DNA-damaging effects of hyperbaric hyperoxia, with the HBO-induced DNA damage consisting largely of single-strand DNA breaks that are rapidly repaired.

## 1. Introduction

There is a growing consensus that high partial pressures of oxygen may induce potentially toxic effects in many cell types and tissues through the reaction of reactive oxygen species (ROS) with biomolecules, particularly the DNA. The production of ROS in excess can be induced by endogenous or exogenous triggers. Examples for exogenous triggers are UV light, radiation [1], pesticides, or toxins [2], or even tobacco, alcohol, or ozone [3]. Physiologically, ROS are constantly produced as a by-product of the respiratory chain in mitochondria [4]. A surplus of oxygen supply and HBO can induce an unphysiological overproduction of ROS, called oxidative stress. In an experimental settings, HBO can, thus, be used to induce oxidative stress [5]. 

Until now, human peripheral blood lymphocytes have been primarily used to examine possible DNA damage after exposure to oxidative stress in human cells, due to their minimally invasive collection [6]. It has not yet been fully investigated if other cell systems react in a similar way to hyperbaric oxygen. In previous studies, our department showed the dose–time dependency between hyperoxia and DNA fragmentation in PBMC and no influence from pressure alone as determined using the comet assay [7], as well as a dose-dependent reduction in DNA damage by the ROS scavengers quercetin and vitamin C in HBO-treated PBMC [8]. Since peripheral lymphocytes are generally in the nonproliferative G0 stage [9], here, we studied the possible implications of HBO on proliferative bone-derived cells. 

Clinical hyperbaric oxygenation therapy (HBOT) is indicated as a treatment for a number of bone diseases, e.g., compromised fracture healing, idiopathic osteonecrosis [10], or in the case of osteoblast hyperproliferation [11]. Most studies concerning osteoblasts and hyperbaric hyperoxia focus on the above topics. Currently, the underlying therapeutic mechanisms are still a focus of research and, moreover, there is only very limited information concerning the potential DNA-damaging effects of hyperoxia on bone cells such as osteoblasts. 

It is expected that tumor cell lines that have an altered metabolism and high proliferation rate will react differently to experimental conditions as compared to primary cells. To take this into account, we used an osteoblastic sarcoma cell line (SAOS-2) in addition to primary human osteoblastic cells (HOB). The high proliferation rate of both cell lines in contrast to, e.g., peripheral lymphocytes, which are mostly in the G0 stage, may sensitize them to oxidative damage but possibly also to faster regeneration [12]. 

For the detection of DNA damage, the alkaline comet assay is established as a sensitive method [13]. Since the alkaline version of this assay is not able to differentiate between single (SSBs) and double-strand breaks (DSBs) [14], we additionally applied a γH2AX focus analysis to stain for γH2AX+53BP1 DSB foci [15] that specifically reveals DSBs [16]. The latter assay is more sensitive and less susceptible to disruptive factors than the comet assay [17], and is able to detect apoptosis [18]. 

In the presence of a surplus of ROS-induced damage, the cell instantly induces antioxidative processes. Hemeoxygenase-1 (HO-1) and NAD(P)H-oxidoreductase-1 (NQO-1), both induced by Nuclear Factor Erythroid-2-related Factor 2 (NrF-2) [19], are two of many genes that are induced by oxidative stress to initiate a variety of antioxidant processes that protect the cell from further oxidative damage [20]. In addition, HO-1 is also known to have antiapoptotic effects and to modulate inflammation and angiogenesis. Thus, a lack of HO-1 is thought to play a role in the pathogenesis of age-related disorders, e.g., cancer or neurodegenerative diseases [21]. In contrast to HO-1 and NQO-1, Transforming Growth Factor beta-1 (TGFß-1) is not a direct marker of oxidative stress, but is an anti-inflammatory cytokine and plays an essential role in the regulation of immune responses [22,23,24]. In addition, TGFß-1 takes part in the complex bone metabolism through controlling the differentiation of osteoblasts and osteoclasts and influencing chemotaxis and proliferation [25,26]. The analysis of the expression of the three mentioned genes is, therefore, one suitable way to examine the possible mechanisms of adaptation of osteoblastic cells to elevated oxidative stress levels. 

To elucidate possible DNA damage after hyperbaric hyperoxia in highly proliferative cells, the present study investigated the induction of DNA damage, apoptosis, and mRNA expression levels of antioxidative marker genes in two osteoblastic cell lines at different time points after hyperbaric hyperoxic exposure.

## 2. Materials and Methods

### 2.1. Cell Lines

The primary human osteoblastic cells (HOB, Promocell^®^, Heidelberg, Germany) and the malignant sarcoma osteoblastic cell line (SAOS-2, DSMZ^®^, Braunschweig, Germany) were used as cell models of osteoblastic cells. HOB cells were cultivated in an osteoblastic growth medium with additional supplement mix (Promocell^®^). For the SAOS-2 cells, McCoy’s 5A medium was supplemented with 15% FCS, 0.5% penicillin–streptomycin, 0.01% NEAA, and 0.01% Glutamax (all Gibco, Paisley, UK). Both cell lines were incubated at 37 °C and 5% CO_2_ and the medium was changed every 72 h. 

### 2.2. Exposure to Hyperbaric Oxygen (HBO)

When cells reached 70–90% confluency in a 75 cm^2^ culture flask, they were resuspended in 12 mL of their respective medium. From the cell suspension, 0.5 mL was transferred to each well of 4 6-well plates and mixed with 2 mL of additional medium per well. Reseeding of the cells was performed 24 h prior to exposure to allow growth to a confluency of at least 60%. Before exposure, cells were harvested from two wells per plate for baseline measurements. Thereafter, two plates were placed in an experimental hyperbaric chamber (Dräger and Haux-Life-Support, D) and exposed to 4.0 bars of 100% O_2_ for four hours at 37 °C. For the control, two plates were placed in a sham chamber and exposed to four hours of ambient air at 37 °C. The cells were harvested immediately before and after exposure as well as after a recovery period of 24 h in an incubator. All analyses for every cell line were performed in ten independent experiments. 

### 2.3. Single-Cell Gel Electrophoresis (Alkaline Comet Assay)

Denaturing single-cell gel electrophoresis was used to detect DNA fragmentation. Cells from one well of the exposed and control plate were harvested and resuspended in 2 mL of culture medium. From this suspension, 50 µL was mixed with low-melting-point (LMP) agarose and then transferred onto microscopic slides covered with medium-electroendosmosis (MEEO) agarose. Slides were then stored in comet assay lysis solution (10 mL of DMSO, 1 mL of Triton-X-100, 89 mL of lysis stock solution, 4 °C, dark) for 24 h, before being subjected to the 12-space electrophoresis chamber (Bauhaus, D). For the denaturation of the DNA, slides were covered with alkaline (pH > 13, 4 °C, dark) buffer for 30 min. The fragmented DNA was then separated with electrophoresis (25 V, 300 mA). For neutralization, the slides were washed 3 times for 5 min with neutralization buffer. After rinsing with dH_2_O, the slides were immersed in 99.8% ethanol and then placed in the dark for 24 h to dry. To visualize the DNA fragments, 50 µL of a 20 µg/mL ethidium bromide solution was applied to each slide. These were then scanned manually with a fluorescence microscope (Olympus, Tokyo, Japan) with a wavelength of 515–560 nm at 400-fold magnification and a band elimination filter of 590 nm. For analysis, a duplicate measurement was used with a target of 200 counted nuclei per slide at each point of measurement. Nuclei were analyzed with a visual binary method (yes/no scoring), which differentiates between obvious damage and no visible damage, and an analysis of the tail moment (the product of tail intensity and length) and tail intensity (the percent of DNA fragmentation in the tail) [27] was performed with Comet Assay IV software. The latter was used as an additional, semiautomatic method as opposed to the visual scoring.

### 2.4. γH2AX Focus Analysis

This analysis was performed to detect DNA double-strand breaks (via γH2AX+53BP1-positive DNA DSB foci analysis) and rate of apoptosis (via pan-nuclear γH2AX staining in the absence of 53BP1 foci). 

Cells from one well of the exposed and control plate were harvested, washed, resuspended in 2 mL of 70% ethanol in PBS, followed by storage at −20 °C. Cells were shipped to Munich were they were analyzed by the team of Harry Scherthan at the Institute of Radiobiology in Munich (for details see [28]).

From the 2 mL suspension, 100 µL (approx. 300,000 cells) was transferred to superfrost^®^ microscopic slides. After centrifugation, the slides were immersed in 1% formaldehyde/PBS solution for 5 min for fixation, followed by rinsing with PBS/0.5% glycine solution for 30 s. The slides were incubated for 10 min in a cuvette (0.25% Triton-X-100/PBS) on ice and then rinsed off three times for five minutes with a PBTG buffer (PBS, 0.1% bovine serum albumin, 0.05% Tween-20, 0.3% fish gelatin, and 37 °C). The slides were subjected to antibody solution (monoclonal anti-γH2AX 1:500 and rabbit-anti-53BP1 1:500, diluted with PBTG buffer), covered with parafilm, and then incubated at 37 °C for 120 min. Again, the slides were rinsed with PBTG buffer then coated with 90 µL of the secondary antibody solution (goat-anti-mouse Alexa-488 1:500 (MoBitec, D); goat-anti-rabbit-Cy3 1:1000 (Dianova, D)) and incubated at 37 °C for 45 min. After the incubation, the slides were washed three times for 5 min with PBTG. An amount of 18 µL of Prolong Gold^®^ antifade solution (Thermo Fisher Scientific, Langenselbold, Germany) was transferred onto the slides and covered with a cover slip. Microscopic slides were recorded as tiled extended focus images using the Metasystems fluorescence microscope imaging system (Metafer 4) equipped with fluorescence filters for red, green, and blue excitation. Extended focus images were exported from the Metafer Imaging System and implemented in the TissueQuest^®^ histology software (TissueGnostics^®^, Vienna, Austria, A) numerical focus analysis. From each sample, the average frequency of foci per cell was computed and the mean and SD were subjected to statistical analysis.

### 2.5. Isolation of RNA and Synthesis of cDNA

Cells from two wells were harvested directly before and after exposure and 24 h after exposure. The suspension was washed with PBS, resuspended in 300 µL of RNA-lysis buffer, and stored at −80 °C. The isolation of RNA was later performed using the Total RNA Kit S Line^®^ from Peqlab Technologies following the manufacturer’s instructions. The concentration of nucleic acid was measured photometrically with the Tecan Infinite M200 pro photometer and accompanying software. The ratio between RNA (wavelength 260 nm) and proteins (wavelength 280 nm) was calculated and samples with ratios over 2.0 were considered pure enough for further usage. 

For the synthesis of cDNA of each sample, 0.1–0.5 µg of RNA, 1 µL of Oligo (dT) primer, and nucleic free water were mixed to 12.5 µL. After a five-minute incubation time at 65 °C, 4 µL of 5x Reaction buffer, 0.5 µL of Ribolock RNAse inhibitor, 2 µL of 10 nM dNTP mix, and 1 µL of RevertAid M-MLV reverse transcriptase (Life Technologies, Carlsbad, CA, USA) were added to each sample to a total volume of 20 µL. The solution was then incubated (42 °C for 60 min, and afterwards, 70 °C for 5 min) before being stored at −20 °C or directly analyzed with real-time PCR. 

### 2.6. Real-Time PCR 

To analyze the expression of the genes of interest (see Appendix A), 2 × 2.5 µL from each cDNA sample were transferred onto a 96-well plate for duplicate measurements. To each well, 7.5 µL of the master mix solution, composed of 5 µL of Light Cycler Sybr Green I Master (Roche Diagnostics, Rotkreuz, Switzerland, CH), 1 µL of primer, and 1.5 µL of water, was added. The PCR was performed using a LightCycler 480 (Roche) for a maximum of 50 cycles including a melting curve analysis as quality control. The results were analyzed with the ∆C(t) method using the following formula: 2^(C(t) GAPDH−C(t) gene of interest)^. A list of the primers used and primer sequences can be found in the Appendix A.

### 2.7. Statistical Analysis 

The data were analyzed using Prism^®^ 8.3. The level of significance was set to ≤0.05. After normality testing with a Kolmogorov–Smirnov test, data from all methods were subjected to a nonparametric Friedman test and post hoc application of Dunn’s multiple comparisons. 

## 3. Results

### 3.1. DNA Fragmentation in HOB and SAOS-2 Cells after Exposure to Hyperbaric Hyperoxia

To examine possible DNA damage after exposure to HBO, the cells were analyzed with an alkaline comet assay. Figure 1 shows the results of the binary “yes/no” scoring in HOB and SAOS-2 cells (A and B), as well as an analysis of the tail moment (C). While no significant changes were observed after exposure to ambient air conditions (sham), a significant increase in DNA fragmentation was observed in both cell lines after four hours of exposure to hyperbaric oxygen compared to baseline (HOB cells: *p* = 0.0003; SAOS-2 cells: *p* = 0.0346) with similar means of absolute damage after exposure (Figure 1A HOB cells 76.02%; SAOS-2 cells 72.16%). Normalized to baseline, the results differed (Figure 1B). Despite similar absolute numbers, HOB cells showed a higher n-fold DNA fragmentation after exposure in comparison to the SAOS-2 cells, with a mean increase of 11.99 times in comparison to a mean increase of 4.27 times in SOAS-2 cells, a 2.8-fold difference. As shown in Figure 1C, the software-assisted analysis of the tail moment confirmed the results of the “yes/no” scoring with a mean 15.21-fold increase in HOB cells and 7.33-fold in SAOS-2 cells after 4 h exposure to hyperbaric hyperoxia compared to baseline. 

### 3.2. DSB Foci Detection and Apoptosis in HOB and SAOS-2 Cells

The alkaline comet assay is known to show DNA SSBs as well as DSBs simultaneously [29]. Thus, to elucidate the type of DNA damage inflicted by HBO exposure more specifically, we next performed a DSB focus analysis by enumerating γH2AX+53BP1-positive DNA DSB foci (Figure 2) [15]. 

The average number of DSBs remained similar in either cell line. Overall, the tumor cell line SAOS-2 showed a higher mean level of DSB damage at all measured points compared to HOB cells (Figure 3A), likely reflecting a higher genomic stress in this tumor cell line. Furthermore, the rate of apoptosis, as measured via pan-nuclear γH2AX staining in the absence of 53BP1 (Figure 3B) [18,28], revealed for both the HOB and SAOS-2 cells a nonsignificant tendency towards an increase in apoptosis rate after 4 h of exposure to HBO. In SAOS-2 cells, a significant increase in apoptotic cells was detected 24 h after exposure (*p* = 0.0421).

### 3.3. Gene Expression of Markers for Oxidative Stress in HOB and SAOS-2 Cells

To analyze the cells’ antioxidative stress response, the mRNA levels of the genes HO-1 and NQO1 were measured using qRT-PCR. In addition, the expression of TGFß-1 as an anti-inflammatory cytokine was measured. The HOB cells showed a significant decrease in TGFß-1 expression after four hours of exposure to hyperbaric oxygen (Figure 4A; *p* = 0.0283, mean basal 8.911 × 10^−5^, 4 h 3.273 × 10^−5^) and a significant increase 24 h after exposure (*p* = 0.0188, mean 24 h 9.220 × 10^−5^). In contrast, the exposure of the SAOS-2 cells to hyperbaric oxygen had no significant influence on the levels of TGFß-1. Instead, a significant increase in expression was seen 24 h after exposure under ambient air (sham) (*p* = 0.002) (Figure 4A). Normalized to baseline, the HOB cells showed a significant mean increase of 1.39 in TGFß-1 24 h after exposure to HBO (*p* = 0.0194) (Figure 5A). 

In contrast to TGFß-1, which was expressed at similar levels in both cell lines, levels of HO-1 and NQO1 were higher in the HOB cells than in the SAOS-2 cells. (HO-1: mean in the HOB cells, 0.01–0.05; in the SAOS-2 cells, 0.001–0.006. NQO1: mean in the HOB cells, 0.12–0.49; in the SAOS-2 cells, 0.05–0.07.) Moreover, in both cell lines, a significant increase in HO-1 expression was measured after 4 h exposure to ambient air (HOB: *p* = 0.0007, mean = 0.05070; SAOS-2: *p* = 0.0007, mean = 0.006445) as well as to HBO in the SAOS-2 cells (SAOS-2: *p* = 0.0012, mean = 0.006275) (Figure 4B). Normalized to baseline (unexposed cells), a significant decrease in HO-1 was shown 24 h after exposure (Figure 5B). While mRNA levels declined in the SAOS-2 cells 24 h after exposure under either condition, HO-1 expression was still elevated 24 h after exposure to HBO in the HOB cells.

Figure 4C shows no significant change in the expression of NQO1 in the SAOS-2 cells, whereas the HOB cells showed a significant increase in NQO1 expression 24 h after exposure to HBO (*p* < 0.0001). Figure 5C shows no significant change in the expression of NQO1 in either cell line when normalized to baseline. 

## 4. Discussion

The focus of this study was the investigation of possible HBO-induced single-strand and double-strand DNA damage and hyperoxia-induced adaptive responses in primary and highly proliferative tumor osteoblastic cell lines. HBO was used as an in vitro method to induce high levels of oxidative stress. 

First, we studied the induction of DNA fragmentation evidenced by SSB formation in the alkaline comet assay after HBO. Our results mirror those of Witte et al. in resting peripheral blood lymphocytes [7]. 

Both osteoblastic cell lines showed a significant increase in DNA fragmentation after exposure to HBO, as well as a decrease in the values 24 h post-HBO treatment, which indicates HBO-induced oxidative stress leads to the DNA damage shown in SSBs and subsequent repair in osteoblasts. Compared to the nonproliferative peripheral lymphocytes in Witte’s study [7], the percentage of damaged nuclei in the proliferative bone cells was increased by more than three-fold by HBO exposure in the benign and malignant osteoblastic cells, which likely relates to cell cycle differences between the cell lines.

Interestingly, the results of the malignant cell line compared to the primary one differed regarding the magnitude of increase in nuclei carrying DNA damage. Normalization of the results to the basal value of nuclei with DNA fragmentation and the tail moment in the comet assay before HBO exposure revealed a significant 12-fold increase in the oxygen-damaged HOB cells, while this increase in the SAOS-2 tumor cells was only 5-fold. This may point to a higher ROS tolerance in the tumor cells. Tumor cell lines generally display increased ROS production [21] and aberrant and unrestrained proliferation [30]. This higher ROS production may be causative for the higher baseline of SSBs in SAOS-2 compared to the HOB cells in our study. While tumor cells may have adapted to the higher ROS levels, the increased proliferation and replication stress may have been picked up by the alkaline comet assay, that also reveals replication-induced DNA breaks [31]. 

Thus, the results of the higher relative increase in the HOB cells, in combination with similar absolute numbers of DNA fragmentation in both cell lines after exposure to hyperbaric oxygen, and the increase in fragmentation relative to exposure to ambient air seem to relate to the relatively low baseline levels of DNA damage in the HOB cells. However, both cell lines showed efficient DNA repair, reflected by the decline in DNA damage 24 h after hyperbaric oxygen exposure. 

A higher amount of DNA damage in the HOB cells after 4 h of sham exposure may relate to external stress factors such as cell handling procedures or the oxygen partial pressure during sham exposure (flow with ambient air), which exceeds physiological standard tissue values [32]. In addition, it has been demonstrated previously that cell culture conditions can induce mild oxidative stress [27] and that mild oxidative stress can stimulate proliferation [33]. Furthermore, the culture medium itself may have influenced the baseline damage, since it has been discussed that cell culture media contain a lower content of antioxidants and partly higher concentrations of, e.g., iron, which can influence the development of oxidative stress through, e.g., the Fenton reaction [34]. 

Since the results of the alkaline comet assay cannot differentiate between the formation of SSBs and DSBs, we also applied the γH2AX+53BP1 foci method to detect DSBs. This in situ stinging method can discriminate apoptotic cells [18,35]. The analysis of DSB-indicating foci [15] revealed that HBO exposure as well as the exposure to ambient air in the experimental chamber failed to induce an increased number of DSB foci in either cell line. An increased baseline level of DSB foci was noted in SAOS-2 cells compared to HOB cells, which aligns with genomic instability and replication stress in tumor cells and agrees with the results of the alkaline comet assay that also showed a lower baseline of DSBs in normal HOB cells.

Taken together, the results of the γH2AX and the alkaline comet assay suggest that the detected HBO-induced DNA damage relates primarily to the formation of SSBs in the DNA of HBO-treated cells. SSBs are likely the consequence of the repair of HBO-induced oxidized bases by base excision repair [36], while elevated proliferation rates in the tumor cell line likely induce replication stress that may also contribute. 

We used the frequency of pan-γH2AX cells to detect apoptosis [28,30] in the differently treated osteoblastic cells. This analysis revealed a nonsignificant increase in apoptotic cells in both cell lines after four hours of exposure to HBO. These findings seem to be in line with the discussed hyperoxia-induced DNA-damaging effects. Accordingly, the significant increase in apoptosis in the SAOS-2 tumor cells after 24 h might be the result of ROS-mediated effects on cells with higher proliferation rates and tumor-specific DNA repair defects and derailed cell cycle regulation [37,38]. Even though previous studies showed hyperoxia-induced apoptosis in tumor cell lines, such as Jurkat T-cells, HL-60, or CCRF-SB cells [39,40,41], we could only detect a tendency in HOB and SAOS-2 cells. Witte’s study, which used nonproliferative peripheral lymphocytes, detected no decrease in cell viability and ATP content [7], which agrees with our results. 

We further analyzed the expression of TGFß-1, HO-1, and NQO-1 genes. While NQO1 [42] and HO-1 [43] are involved in the oxidative stress response, HO-1 also has antiapoptotic effects [44]. TGFß-1, besides its well-known effects on tumor cells, is involved in regulating immune responses [26,45].

The gene expression of TGFß-1 was slightly negatively affected in both cell lines after HBO. TGFß-1 as a cytokine may mediate anti-inflammatory effects by regulating immune responses [26]. The mechanisms behind the positive effects of (hyperbaric) hyperoxia, e.g., on wound healing, are still not fully understood. One possible explanation is an immunomodulatory effect, since studies have shown an influence on different cytokines and other immune parameters [46,47]. In contrast, Kiers et al. have more recently shown that the positive effects are more likely due to better oxygenation than direct immunomodulatory qualities [48]. Thus, it is not clear yet if the affected TGFß-1 expression does result in an immunomodulatory effect, and further investigations are needed. Oxidative stress, as induced in our experiments, seems to have an upregulating effect on the expression of TGFß-1, at least in SAOS-2 cells, which showed a significant increase in expression 24 h after sham exposure. A possible induction of TGFß-1 by mild stress through, e.g., cell culture as discussed above is also consistent with its function as a proliferation inducer in osteoblasts and the understanding that mild oxidative stress can stimulate proliferation [33,49]. 

HO-1 is known to be increased in oxidative stress, plays a role in antioxidative protection and adaptation to oxidative stress, and has antiapoptotic effects [43,44]. The significant increase in HO-1 expression in both cell lines already after 4 h of exposure to ambient air again points towards an induction of mild oxidative stress under sham conditions in this study. In addition, the difference in the pattern of HO-1 increase in the HOB cells between the control and experimental pressure chambers is of interest: mild oxidative stress seems to have had only a short-term inducing effect on HO-1 expression, as sham exposure with ambient air led to a significant drop in HO-1 expression after 24 h. On the other hand, hyperbaric oxygen exposure induced a slight increase up to 24 h in the HOB cells.

Similar findings were also obtained by Speit in the A549 lung tumor cell line [50], who observed an increase in HO-1 expression 24 h after hyperbaric oxygenation, while a second exposure to HBO resulted in significantly less damage in the comet assay pointing to an adaptive response [50]. The decrease in damage after 24 h was more likely due to fast base excision repair, since we assume that, as discussed previously, DNA damage relates primarily to the formation of single-strand breaks [51]. Even though the induction of HO-1 as part of antioxidative adaptive processes might play a role, the results suggest that no significant cell adaptation was induced. This is in line with the fact that the level of mRNA expression usually shows an earlier fold change than the actual protein [52]. Thus, the results might not be biologically relevant, even though the analysis showed significant changes. 

NQO1, like HO-1, is induced through the activation of Nrf2 as a protective response to oxidative stress [53]. In the HOB cells, an expression response to HBO was evident from the significant increase in NQO1 expression 24 h after HBO exposure, and this is in good correlation with our findings on HO-1 expression. Thus, the cellular response to oxidative stress was similar in the expression of HO-1 and NQO1, and appears to function similarly in osteoblasts as in lymphocytes or lung cells [50,54,55]. However, the expression of NQO1 in the SAOS-2 cells was not affected by HBO exposure and remained at very low levels at all time points. Although other studies have demonstrated NQO1 expression in SAOS-2 cells [56] and NQO1 to be an early-response gene to oxidative stress [57], our study indicates that its expression is not impacted by oxidative stress caused by exposure to hyperbaric oxygenation in this malignant cell line. 

## 5. Conclusions

Overall, the similar results in both cell lines support the findings that hyperbaric oxygenation also has a significant influence on the development of oxidative DNA damage in osteoblastic cells, and that the repair and adaptation patterns triggered are similar to other cell systems. The findings of this study suggest that, in comparison to nonproliferative leukocytes [7], highly proliferative osteoblastic cells are a lot more susceptible to DNA damage through HBO than nonproliferative cells. In addition, the results of the comet assay and γH2AX+53BP1 DSB-indicating foci suggest that both osteoblastic cells primarily suffer SSBs rather than DSBs after HBO stress and are able to prevent long-term DNA damage, likely through efficient DNA repair. Thus, the findings of this study are in agreement with the described positive effects of oxygen pressure therapy in bone diseases [58,59,60]. 

## Figures and Tables

**Figure 1 antioxidants-12-00851-f001:**
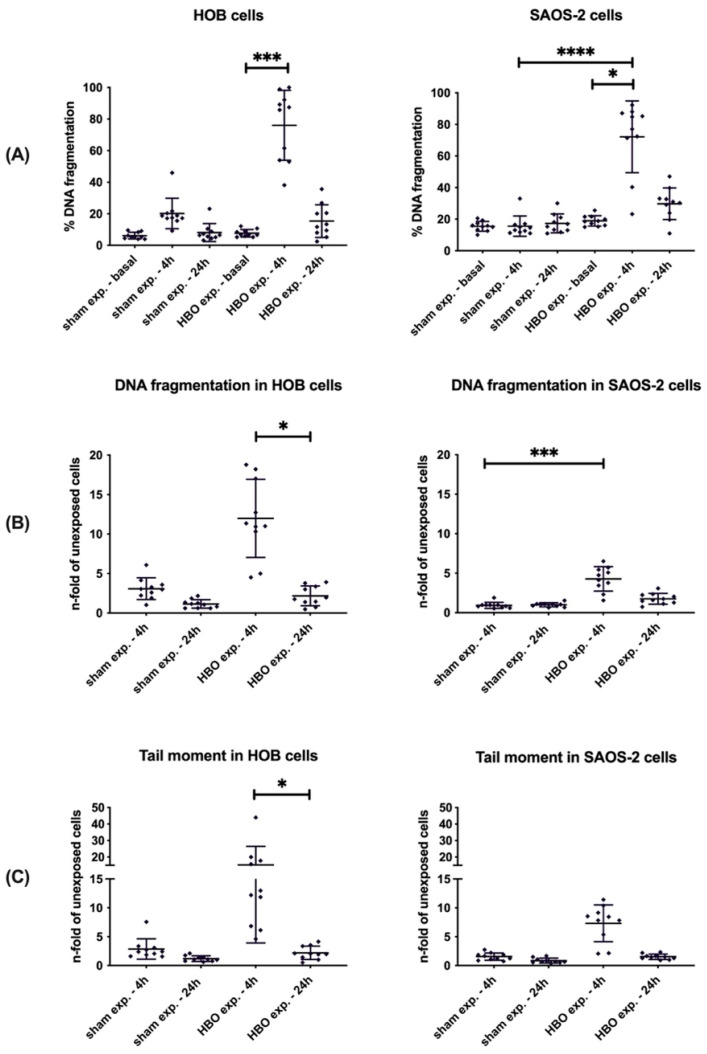
DNA fragmentation in human osteoblasts after exposure to hyperbaric oxygen (HBO exposure) and ambient air (sham exposure) analyzed with alkaline comet assay. (**A**) % DNA fragmentation in HOB and SAOS-2 cells; (**B**) DNA fragmentation in HOB and SAOS-2 cells after normalization to basal value; (**C**) DNA damage in HOB and SAOS-2 cells shown as tail moment after normalization to basal value. Data are presented as mean ± SD; *p*-values: * *p* < 0.05, *** *p* < 0.001, **** *p* < 0.0001.

**Figure 2 antioxidants-12-00851-f002:**
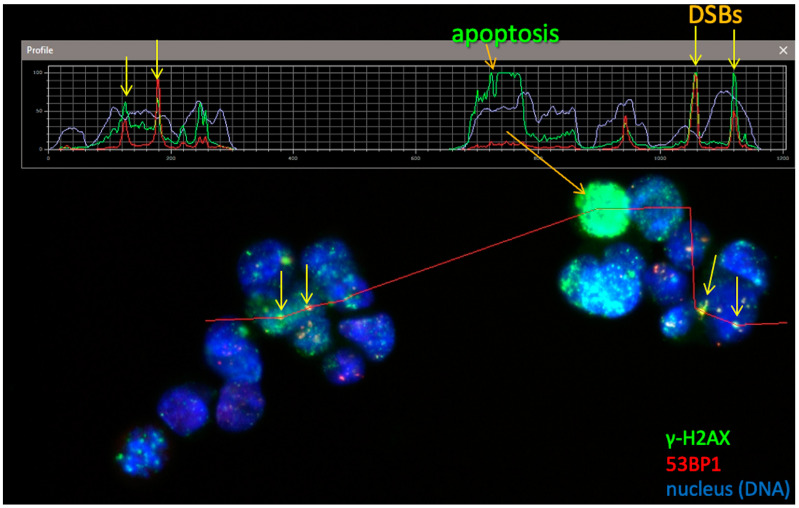
DSB focus analysis in HOB cells via γH2AX+53BP1-positive DNA DSB foci detection. The upper part of the image shows intensity profiles along the red line below across 6 nuclei (DNA, blue). γH2AX-positive foci are shown in green, 53BP1 in red. Colocalized foci are a surrogate marker for DSBs and are marked with yellow arrows. Apoptosis can be seen as pan-nuclear γH2AX staining in absence of 53BP1.

**Figure 3 antioxidants-12-00851-f003:**
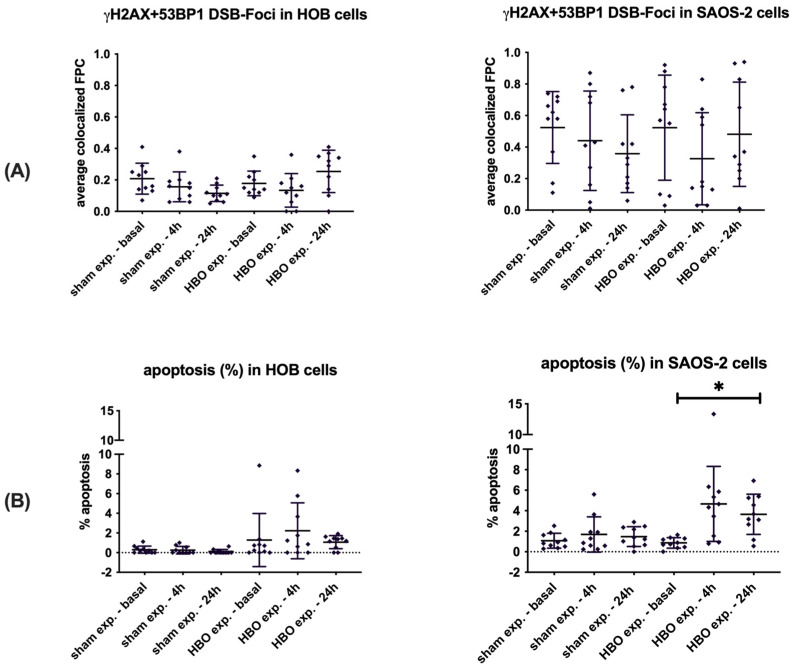
Detection of γH2AX+53BP1 double-strand break foci (**A**) and apoptosis (**B**) in HOB and SAOS-2 cells after exposure to hyperbaric oxygen (HBO exposure) and ambient air (sham exposure). Data are presented as mean ± SD; *p*-value: * *p* < 0.05.

**Figure 4 antioxidants-12-00851-f004:**
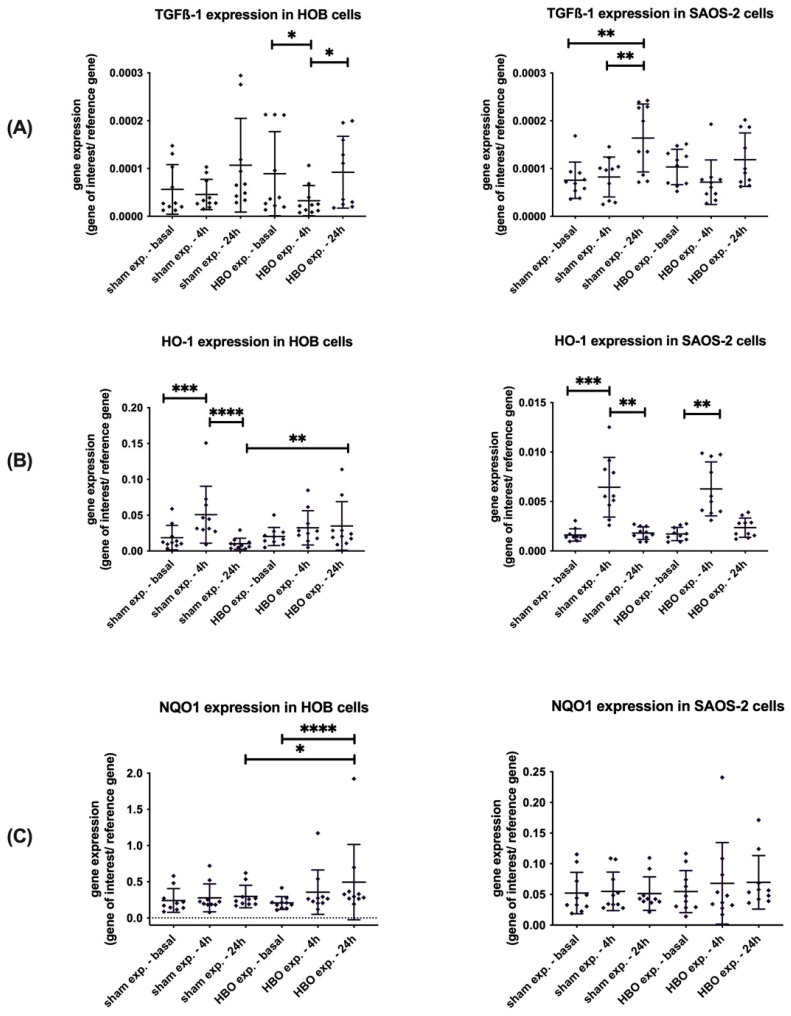
Gene expression of TGFß-1 (**A**), HO-1 (**B**), and NQO1 (**C**) in HOB and SAOS-2 cells after exposure to hyperbaric oxygen (HBO exposure) and ambient air (sham exposure). Data are presented as mean ± SD; *p*-values: * *p* < 0.05, ** *p* < 0.01, *** *p* < 0.001, **** *p* < 0.0001.

**Figure 5 antioxidants-12-00851-f005:**
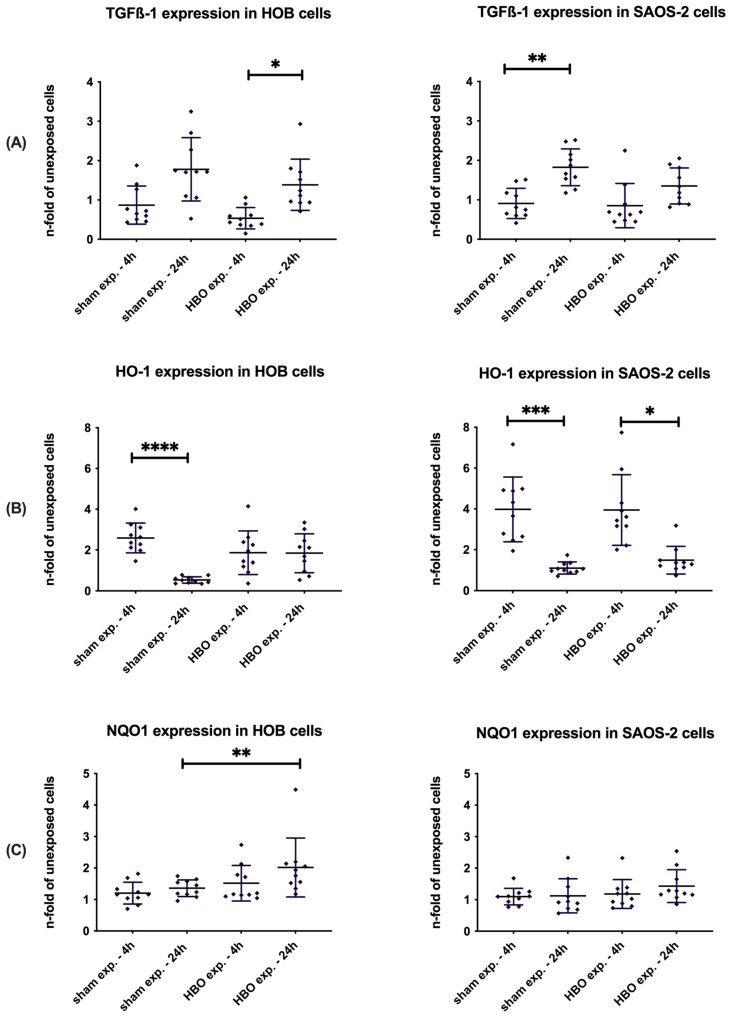
Gene expression of TGFß-1 (**A**), HO-1 (**B**), and NQO1 (**C**) in HOB and SAOS-2 cells after exposure to hyperbaric oxygen (HBO exposure) and ambient air (sham exposure) normalized to baseline. Data are presented as mean ± SD; *p*-values: * *p* < 0.05, ** *p* < 0.01, *** *p* < 0.001, **** *p* < 0.0001.

## Data Availability

Data is contained within the article and Appendix A.

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
