# Peer review of "Analysis of Single- and Double-Stranded DNA Damage in Osteoblastic Cells after Hyperbaric Oxygen Exposure"

_antioxidants, 2023, doi:10.3390/antiox12040851_

Round 1

Reviewer 1 Report

The Authors have investigated the effects of hyperbaric oxygen (HBO) in vitro in proliferating bone cells (primary osteoblasts HOB and osteogenic tumor cell line SAOS-2).  The aim of the study was to evaluate the DNA damaging effect of the generation of oxidative stress, known to be associated with HBO therapy, in bone cells. They also evaluated the expression of genes associated with antioxidant responses (HO-1 and NQO1) and tissue repair (TGFbeta).

They show a significant induction of DNA damage in both cell lines, mainly represented by single strand breaks (SSBs), which are easily repaired, at variance with double strand breaks (DSBs). Indeed they also show that this damage peaks at 4h and is no longer detected at 24. This is of course good news for HBO therapy applications.

Points to be modified:

- The Authors did not find major changes induced by HBO in gene expression: both for TGFbeta and HO-1, similar changes were found in sham and experimental conditions over time. This further confirms that HBO does not induce significant damage to the cells and thus does not require significant cell adaptation. The sentence "But it might be assumed, that the induction of HO-1 as part of anti-oxidative adaptive processes plays a role, too." (lines 378-379) should be modified. Same for line 401 "induction of adaptation processes".

- Lines 285-286: "HBO-induced oxidative stress leads to base oxidations in DNA shown in SSBs and subsequent repair in osteoblasts." This sentence should be modified: the authors do not have experimental data on base oxidation, and oxidative stress can directly induce SSBs, which do not require BER o NER to be repaired. Same for lines 393-397 "hyperbaric oxygenation also has a significant influence on the development of oxidative DNA base damage in osteoblastic cells and that the repair and adaption patterns triggered are similar to other cell systems. The findings of this study suggest in comparison to non-proliferative leukocytes [7], that highly proliferative osteoblastic cells are a lot more susceptible to base oxidation through HBO than non-proliferative cells."

- Change "tumorous" to "cancer" or "tumor" throughout the paper

Reviewer 2 Report

Please add experiments with 4ATA of air.  The damage may be caused not only by 100% oxygen but also by high pressure.

Please write the data of equipment.

Table 1 is suitable for supplemental data.

Reviewer 3 Report

In this manuscript the authors examined effect of hyperbaric oxygen on the genome on osteoblastic cells.

While interesting this manuscript can be improved in several areas:

First: The authors must do a better job of editing and proof-reading before submitting. They overlooked that references were not properly entered into the manuscript.

Second: Figures are not properly labeled and some references to figures are missing in manuscript.

Third: The manuscript is missing some key experiments: Is the DNA damage observed after HBO due to reactive oxygen? Does NAC alleviate the DNA damage? How does HBO affect cell cycle progression? Are all the cells demonstrating increased 53BP1/gH2AX foci in S-phase or is this independent of cell-cycle phase. Lasty, why didnt the authors examine double strand breaks directly in combination with 53BP1/gH2AX foci. This can be easily done by using a neutral comet assay, which is basically the same protocol as the alkaline comet assay. Omission of this experiment is a major concern.

The manuscript appears to be hastily put together and data is very preliminary.

Round 2

Reviewer 1 Report

Authors have complied with all the criticisms raised by this reviewer.

Author Response

Thank you for the constructive review. 

Reviewer 2 Report

Please show the importance of Table 1 in the manuscript, such as ingenuity about the conditions of qRT-PCR.

Please check CO2 and O2 (subscript).

Reviewer 3 Report

It appears that the authors have addressed the concerns from the previous review. There are no concerns with this manuscript

Author Response

Thank you for the constructive review